# Knowledge of Human Monkeypox and Its Relation to Conspiracy Beliefs among Students in Jordanian Health Schools: Filling the Knowledge Gap on Emerging Zoonotic Viruses

**DOI:** 10.3390/medicina58070924

**Published:** 2022-07-11

**Authors:** Malik Sallam, Kholoud Al-Mahzoum, Latefa Ali Dardas, Ala’a B. Al-Tammemi, Laith Al-Majali, Hala Al-Naimat, Laila Jardaneh, Farah AlHadidi, Khaled Al-Salahat, Eyad Al-Ajlouni, Nadin Mohammad AlHadidi, Faris G. Bakri, Azmi Mahafzah, Harapan Harapan

**Affiliations:** 1Department of Pathology, Microbiology and Forensic Medicine, School of Medicine, The University of Jordan, Amman 11942, Jordan; khalid_1986_6@yahoo.com (K.A.-S.); aya8191774@ju.edu.jo (E.A.-A.); mahafzaa@ju.edu.jo (A.M.); 2Department of Clinical Laboratories and Forensic Medicine, Jordan University Hospital, Amman 11942, Jordan; nadeen.hadidi@yahoo.com; 3Department of Translational Medicine, Faculty of Medicine, Lund University, 22184 Malmö, Sweden; 4School of Medicine, The University of Jordan, Amman 11942, Jordan; kaalhajeri85@icloud.com (K.A.-M.); lyt0209361@ju.edu.jo (L.A.-M.); hla0201058@ju.edu.jo (H.A.-N.); frh0205722@ju.edu.jo (F.A.); 5School of Nursing, The University of Jordan, Amman 11942, Jordan; l.dardas@ju.edu.jo; 6Migration Health Division, International Organization for Migration (IOM), The UN Migration Agency, Amman 11953, Jordan; dr.alaatamimi@yahoo.com; 7School of Dentistry, The University of Jordan, Amman 11942, Jordan; lyl2190611@ju.edu.jo; 8Department of Internal Medicine, School of Medicine, The University of Jordan, Amman 11942, Jordan; fbakri@ju.edu.jo; 9Infectious Diseases and Vaccine Center, The University of Jordan, Amman 11942, Jordan; 10Medical Research Unit, School of Medicine, Universitas Syiah Kuala, Banda Aceh 23111, Indonesia; harapan@unsyiah.ac.id; 11Tropical Disease Centre, School of Medicine, Universitas Syiah Kuala, Banda Aceh 23111, Indonesia; 12Department of Microbiology, School of Medicine, Universitas Syiah Kuala, Banda Aceh 23111, Indonesia

**Keywords:** attitude, rumors, preventive behavior, conspiracy theory, biological warfare, lockdown, Orthopoxviruses

## Abstract

*Background and Objective:* The recent multi-country outbreak of human monkeypox (HMPX) in non-endemic regions poses an emerging public health concern. University students in health schools/faculties represent a core knowledgeable group that can be helpful to study from a public health point of view. As future healthcare workers, assessment of their knowledge and attitude towards emerging zoonotic viral infections can be helpful to assess their taught material and courses with potential improvement if gaps in knowledge were identified. Therefore, we aimed to evaluate the level of HMPX knowledge, conspiracy beliefs regarding emerging virus infections, as well as their associated determinants among university students studying Medicine, Nursing, Dentistry, Pharmacy, Medical Laboratory Sciences, and Rehabilitation in Jordanian health schools/faculties. In addition, we sought to evaluate the correlation between HMPX knowledge and the extent of holding conspiracy beliefs regarding emerging viral infection. *Materials and Methods:* A convenient sample of university students was obtained through an electronic survey distributed in late May 2022 using the chain-referral approach. Assessment of HMPX knowledge and general attitude towards emerging virus infections was based on survey items adopted from previously published literature. *Results:* The study sample comprised 615 students with a mean age of 20 years and a majority of females (432, 70.2%) and medical students (*n* = 351, 57.1%). Out of eleven monkeypox knowledge items, three were identified correctly by >70% of the respondents. Only 26.2% of the respondents (*n* = 161) knew that vaccination to prevent monkeypox is available. Age was significantly associated with better HMPX knowledge for a majority of items. Older age, females, and affiliation to non-medical schools/faculties were associated with harboring higher levels of conspiracy beliefs regarding emerging virus infections. Our data also indicate that lower levels of HMPX knowledge were associated with higher levels of conspiracy beliefs. *Conclusion:* The current study pointed to generally unsatisfactory levels of knowledge regarding the emerging HMPX among university students in Jordanian health schools/faculties. Conspiracy beliefs regarding emerging virus infections were widely prevalent, and its potential detrimental impact on health behavior should be evaluated in future studies.

## 1. Introduction

Human monkeypox (HMPX) is a zoonotic disease that has been described for more than 60 years, with the first recorded human case dating back to 1970 [1,2]. The causative agent is the monkeypox virus (MPXV), which is classified in the genus *Orthopoxvirus* within the family *Poxviridae* [3].

Since the successful eradication of the notorious variola virus infection (smallpox) from the human population in late 1970s, MPXV emerged as an important threat among other *Orthopoxvirus* members [4]. This threat was manifested in several outbreaks of the disease, mostly in Western and Central Africa, with cases in the United States (U.S.) and Europe that were associated with imported animals or a travel history to endemic areas [2,5,6,7]. The potential threat of HMPX materialized in 2022 with an incremental increase in cases in non-endemic regions [8]. The World Health Organization (WHO, Geneva, Switzerland) described the status of HMPX as a multi-country outbreak in non-endemic countries constituting a moderate public health risk at the global level [9].

The transmission of HMPX occurs mostly via the respiratory tract/saliva or by direct contact with skin lesions of the infected animals [10]. Human-to-human transmission has been reported prior to the current outbreak mostly among household contacts or in hospital settings [11,12]. However, human-to-human transmission did not appear to occur as readily as in smallpox, with household attack rates of 3–11% [13]. During the current HMPX outbreak, the predominance of cases among men having sex with men (MSM) raises questions regarding the possible sexual transmission of the virus [14,15]. However, the clustering of cases among MSM can be due to the founder effect following the introduction of MPXV into MSM with subsequent transmission through close contact [16].

Regarding the clinical presentation, HMPX and smallpox share similar signs and symptoms, albeit milder with better outcome [17]. For HMPX, a prodromal phase of fever, headache, fatigue, and lymphadenopathy ensues following an incubation period of 7–17 days, which may also range from 5 to 21 days [18,19]. Declining fever accompanies the eruption of the centrifugally distributed skin rash that evolves through macular, papular, vesicular, and pustular phases, with each phase lasting 1–2 days [18]. The skin rash was reported to be more apparent on the face and limbs than on the trunk [20].

Antivirals can be used to treat HMPX [21]; nevertheless, the currently used drugs (brincidofovir and tecovirimat) have been licensed for the treatment of smallpox rather than HMPX [22,23]. The prevention of HMPX relies on smallpox vaccination, which has been reported to provide a protection level of about 85% [24,25]. In line with this cross-protection, the increased susceptibility to monkeypox following the cessation of smallpox vaccination can be explained by the decrease in immunity previously conferred by smallpox vaccination [26,27].

The epidemiology of MPXV infection has been dominated by two phylogenetically distinct clades (monophyletic taxa descending from a common ancestor) of the virus [28]. The West African clade and the Central African (Congo Basin) clade are characterized by noticeable differences in terms of geographic distribution, case-fatality ratio (CFR), and transmission. The West African clade has been linked to a CFR of 3%, while the Congo Basin clade showed a CFR of 11% [29]. Phylogenetic analyses of a few isolates during the current outbreak (from Belgium and Portugal) revealed the presence of the Western African clade of MPXV; however, more comprehensive molecular studies are warranted to reveal the phylodynamic characteristics of the ongoing HMPX outbreak [30,31].

An important point to be emphasized is the urgent need to adopt a new non-discriminatory and non-stigmatizing nomenclature scheme for MPXV clades [32]. One proposed classification scheme was conceptualized by Happi et al. based on maximum likelihood phylogenetic analysis of the available MPXV genomes (1970–2022). This “Happi” classification system suggested the adoption of Arabic numerals to assign MPXV clades based on their order of detection rather than source of isolation [32]. Thus, the older nomenclature will be rendered obsolete by the use of “MPXV clade 1” instead of Congo Basin clade, and “MPXV clades 2 and 3” instead of West African clade, with the newly assigned clade 3 incorporating most genomes from the human outbreaks that were recorded in 2017, 2018, and the ongoing 2022 multi-country outbreak. Additionally, the proposed “Happi” scheme inferred the discernible genetic diversity of the taxa within clade 3; therefore, the adoption of the “Pango” nomenclature scheme used for SARS-CoV-2 was suggested for lineages within this clade (e.g., A.1, A.2, A.1.1, and B.1 representing the taxa that were sequenced during the ongoing multi-country HMPX outbreak) [32,33].

Previously, the WHO stated that one of the challenges to prevent the reemergence of HMPX could be the lack of knowledge of the disease [34]. One of the challenges to control the ongoing outbreak is the wide prevalence and rapid spread of rumors about the disease. The spread of rumors and misinformation online poses risks that were noticed worldwide, including Arab countries, and revolved around the following: (1) coronavirus disease 2019 (COVID-19) vaccination is linked to the outbreak, (2) Microsoft co-founder and billionaire Bill Gates has a role in the outbreak, and (3) governments falsely augment the fears about the disease [35,36].

Tackling knowledge can be considered the first step in the attempt to change attitudes and behavior. Mounting knowledge is often associated with a greater influence of attitudes on behavior. That is, when attitudes are grounded in high amounts of knowledge, they are more enduring and consequential and better predictors of behaviors than when they are based on little or false knowledge [37,38].

Assessment of knowledge levels among students in health schools to emerging viral infections can be relevant in terms of their preparedness as future healthcare workers and willingness to work during outbreaks of infectious diseases [39,40]. In addition, the assessment of the relation between disease knowledge and attitude towards conspiracy beliefs can have implications on the understanding of health-seeking behavior, including the likelihood to adhere to preventive measures such as vaccination [41,42].

Furthermore, university students rely heavily on social media and thus, their knowledge is at particular risk for being contaminated with online rumors and conspiracies [43,44]. Consequently, this can result in less engagement in the protective behavior and the preventive efforts aiming to tackle infectious diseases, as shown recently by Valerie van Mulukom amid the COVID-19 pandemic [45,46].

The overarching goal of the current study was to evaluate the basic level of HMPX knowledge among university students in health schools/faculties in Jordan. The relevance of conducting such research in Jordan is based on the previous evidence of a wide prevalence of conspiracy beliefs and circulating misinformation that was shown in the country during the COVID-19 pandemic [44,45,47,48,49].

The specific aims were to: (1) assess possible variables that could be associated with higher levels of HMPX knowledge and (2) to explore the potential correlation between HMPX knowledge and harboring conspiratorial beliefs regarding emerging virus infections. This objective was based on the recurrent pattern of circulating rumors and unfounded claims about the origin of HMPX [50].

## 2. Materials and Methods

### 2.1. Study Design and Setting

The current study was based on a cross-sectional design. The distribution of the online questionnaire was conducted between 24 May 2022 (21:00) and 26 May 2022 (23:59). The inclusion criteria were: (1) age ≥ 18 years, (2) current enrolment in Jordanian universities/colleges, and (3) affiliation to one of the following health schools/faculties: Medicine, Dentistry, Pharmacy, Nursing, Rehabilitation, and Medical Laboratory Sciences.

Recruitment of the potential respondents was performed through chain-referral sampling, starting with the contacts of the authors (four of whom are instructors of Medical, Dental, Nursing, and Medical Laboratory Sciences students, and five Medical/Dental students at the University of Jordan) with reliance on participants’ referral of the survey link to their contacts [51]. The survey link was created in Google Forms, and the link was shared on the following social media platforms/free messaging services: Facebook, Instagram, WhatsApp, and Telegram.

The survey was distributed in Arabic language without incentives for participation. Response to all items was mandatory to overcome the item non-response issue.

Calculation of the sample size was based on the currently available data pointing to about 50,000 university/college students in health schools/faculties in Jordan (personal communication, the Ministry of Higher Education, Amman, Jordan). Thus, the minimum required sample size was 594 based on a 95% confidence interval and a 4% margin of error [52].

### 2.2. Ethical Permission

The current study was approved by the Scientific Research Committee at the School of Medicine/University of Jordan on 24 May 2022 (reference number: 2544/2022/67). An informed electronic consent was obtained from the respondents using the following item in the introductory section of the survey: “Do you agree to participate in this study?”. The respondent had to respond “Yes” to be able to participate in this study.

### 2.3. Overview of the Questionnaire and Study Variables

There were two response variables in this study: (1) knowledge on HMPX and (2) conspiracy beliefs regarding emerging virus infections.

The first section of the survey following the informed consent section involves items to assess respondent age, sex, residence (the capital Amman vs. outside the capital), and school/faculty (Medicine, Dentistry, Pharmacy, Nursing, Medical Laboratory Sciences, or Rehabilitation).

The survey items assessing the level of monkeypox knowledge were adopted from Harapan et al. [53]. The possible responses to each knowledge item were (yes vs. no vs. I do not know). Correct responses were scored as 1, incorrect responses were scored as −1, and “I do not know” was given a score of zero, which were used as a sum to represent the monkeypox knowledge score (MPX K-score).

Monkeypox knowledge was assessed using an 11-item section as follows: (1) Monkeypox is prevalent in the Middle East (incorrect), (2) Monkeypox is prevalent in Western and Central Africa (correct), (3) There is an outbreak of human monkeypox in the world (correct), (4) Monkeypox is caused by a virus (correct), (5) Human-to-human transmission of monkeypox occurs easily (incorrect), (6) Monkeypox and smallpox have similar signs and symptoms (correct), (7) Skin rash is one of the signs or symptoms of human monkeypox (correct), (8) Pustule is one of the signs or symptoms of human monkeypox (correct), (9) Antibiotics are used to treat human monkeypox (incorrect), (10) Diarrhea is one of the signs or symptoms of human monkeypox (incorrect), and (11) Vaccination is available to prevent human monkeypox (correct). Poor level of knowledge per item and for the overall assessment was defined at a 70% correct responses level [53].

Regarding the assessment of attitude towards conspiracy explanations of emerging virus infections, we adopted survey items from a study by Freeman et al. on coronavirus conspiracy beliefs [54]. The evaluation was carried out through a 12-item section, with 7-Likert scale possible responses (strongly disagree (1), disagree (2), somewhat disagree (3), neutral/no opinion (4), somewhat agree (5), agree (6), strongly agree (7)).

The following items comprised the emerging virus infections conspiracy scale (EVICS): (1) “I am skeptical about the official explanation regarding the cause of virus emergence”, (2) “I do not trust the information about the viruses from scientific experts”, (3) “Most viruses are man-made”, (4) “The spread of viruses is a deliberate attempt to reduce the size of the global population”, (5) “The spread of viruses is a deliberate attempt by governments to gain political control”, (6) “The spread of viruses is a deliberate attempt by global companies to take control”, (7) “Lockdowns in response to emerging infection are aimed for mass surveillance and to control every aspect of our lives”, (8) “Lockdowns in response to emerging infection are aimed for mass surveillance and to destabilize the economy for financial gain”, (9) “Lockdown is a way to terrify, isolate, and demoralize a society as a whole in order to reshape society to fit specific interests”, (10) “Viruses are biological weapons manufactured by the superpowers to take global control”, (11) “Coronavirus was a plot by globalists to destroy religion by banning gatherings”, and (12) “The mainstream media is deliberately feeding us misinformation about the virus and lockdown” [54].

Higher EVICS scores indicated a higher embrace of conspiracy beliefs regarding virus emergence and subsequent intervention measures. To assess the relevance and representativeness of the EVICS and the knowledge items, content validity was checked by the first and the senior authors (M.S., A.M. and H.H.). The internal consistency of EVICS was ensured by a Cronbach’s alpha value of 0.930.

We collected and included some possible explanatory variables in this study: age, sex, place of residence, and type of school/faculty. Age of the respondents was divided into two groups (<21 years and ≥21 years), based on the median age of 20 years in the study sample. The residency of the respondents was grouped into those residents in the capital city of Amman and outside Amman. Type of school/faculty was divided into Medicine, Dentistry, Nursing, Pharmacy, Laboratory Sciences, and Rehabilitation. However, for analysis purposes, the type of school/faculty was divided into Medical and Non-Medical “Others” (Dentistry, Nursing, Pharmacy, Laboratory Sciences, and Rehabilitation) based on the relatively low number of respondents affiliated with non-medical schools/faculties and the intensive coverage of infectious disease topics in medical schools compared to non-medical schools.

### 2.4. Statistical Analysis

The statistical analyses were conducted through IBM Statistical Package for the Social Sciences (SPSS) for Windows, Version 22.0. Armonk, NY: IBM Corp.

The characterization of the scale variables was based on measures of central tendency (mean) and dispersion (standard deviation (SD)). To test for normality of distribution of the scale variables (age, MPX K-score, and EVICS), we used the Kolmogorov–Smirnov test (K-S) considering the relatively large sample size in this study.

Associations of explanatory variables and HMPX knowledge or conspiracy beliefs regarding emerging virus infections were evaluated using the chi-squared test (χ^2^) or the two-tailed Mann–Whitney *U* test (M-W) as appropriate. Univariate regression analysis was used as appropriate. The statistical significance was set at *p* < 0.050 as the cut-off level.

## 3. Results

### 3.1. Characteristics of the Study Sample

The total number of study respondents that comprised the final sample was 615 students. The general characteristics of the study respondents are illustrated in Table 1. Medical and dental students prevailed in the study sample, while female students represented the majority of respondents across all schools/faculties (63.2% of medical students, 75.0% of nursing students, 78.0% of dental students, 82.6% of pharmacy students, 85.7% of rehabilitation students, and 87.1% of laboratory sciences students).

To check for normal distribution of age, non-normality was found (*p* < 0.001, K-S; skewness = 2.408, kurtosis = 15.320). This forced the use of non-parametric tests (M-W) for the assessment of possible correlations between age and other categorical variables. Male students had a lower mean age compared to females (19.7 vs. 20.0 years, *p* = 0.036, M-W). Medical students had a lower mean age as well, compared to students in other schools/faculties (19.7 vs. 20.2 years, *p* < 0.001, M-W).

The percentage of male medical students was significantly higher compared to the percentage of male respondents from other schools/faculties (36.8% vs. 20.5%, *p* < 0.001, χ^2^ = 19.147). The place of residence among the study respondents did not significantly differ based on sex, age, and faculties/schools.

### 3.2. Human Monkeypox Knowledge and Associated Determinants

The overall level of knowledge regarding human monkeypox was poor, with only three items having correct response levels > 70% (Figure 1). Notably, only 26.2% (*n* = 161) of the respondents were aware of the presence of vaccination to prevent human monkeypox.

Statistically significant differences in the level of monkeypox knowledge were observed for a majority of the items between medical and non-medical students (9/11). Non-medical students had a higher level of knowledge compared to medical students for eight items, compared to a single item where medical students displayed a significantly higher level of knowledge (Figure 2).

However, the assessment of the difference between the mean MPX K-score among medical vs. non-medical students did not yield a statistically significant difference (4.2 vs. 4.1, respectively, *p* = 0.523, M-W). The mean MPX K-score was similar upon stratification per sex (4.2 for both male and female students, *p* = 0.889, M-W).

The differences were less conspicuous upon comparing the level of knowledge based on sex. However, age appeared to have a significant association, with better knowledge among older participants (Table 2).

### 3.3. Conspiracy Beliefs regarding Emergence of Virus Infections and Its Associated Determinants

Our analysis indicated that the EVICS score did not distribute normally (*p* < 0.001, K-S; skewness = 0.138, kurtosis = −0.630). This forced the use of the non-parametric tests (M-W) for the assessment of possible correlations between EVICS and categorical explanatory variables.

Regarding the item “Viruses are biological weapons manufactured by the superpowers to take global control”, 50.5% of the study respondents agreed at least to some extent with this claim (Figure 3).

Higher mean EVICS score indicating a higher embrace of conspiracy beliefs regarding virus infection emergence was found among females (45.0 vs. 39.8, *p* < 0.001, M-W). Lower mean EVICS score was found among medical students compared to those in other health schools/faculties (41.5 vs. 46.0, *p* < 0.001, M-W, Figure 4). Respondents younger than 21 years old had a higher mean EVICS score compared to those aged 21 or older (44.6 vs. 40.1, *p* = 0.001, M-W, Figure 4). The place of residence (Amman vs. outside Amman) did not show any significant differences in EVICS score (43.1 vs. 44.8, *p* = 0.386, M-W).

Based on the mean and median monkeypox K-score (4.2, 4.0), the study sample was divided into two group: better knowledge (MPX K-Score > 4) and inferior knowledge (MPX K-Score ≤ 4). Those with better knowledge had significantly lower mean EVICS scores (41.4 vs. 45.2, *p* = 0.002, M-W, Figure 4).

Additionally, univariate analysis with the EVICS score as the dependent variable, MPX K-score dichotomized into better and inferior as the fixed factor, and the following as covariates: sex, school/faculty, and age, showed that a higher MPX K-score was associated with a lower embrace of conspiracy beliefs regarding virus emergence (*p* = 0.009), with age (*p* = 0.001), school/faculty (*p* = 0.014), and sex (*p* = 0.019) as significant covariates.

## 4. Discussion

The ongoing outbreak of HMPX in several countries worldwide has brought into focus the issue of conspiracies regarding emerging virus infections [55]. The belief in conspiratorial ideas, particularly those involving health-related topics, is widespread [56]. Conspiratorial beliefs can result in negative consequences by abstaining from adhering to appropriate health behaviors among those endorsing such beliefs [45,57,58].

The widespread prevalence of conspiracy beliefs was manifested amid the COVID-19 pandemic with suggested negative psychological, social, and health impacts [59,60,61]. Specifically, our previous research has shown that the belief in the manmade origin of severe acute respiratory syndrome coronavirus 2 (SARS-CoV-2), and the endorsement of the notion that COVID-19 is part of a biologic warfare, was associated with higher anxiety levels among the general public and among university students in Jordan [47,48].

In addition, recent studies in Jordan showed that the conspiratorial thinking was linked to negative impacts on health-seeking behavior in term of its correlation with COVID-19 vaccine hesitancy among the general public and university students in the country [44,49]. Therefore, we aimed to assess the extent of endorsement of conspiracy beliefs regarding emerging virus infections through the proposed EVICS scale. This aim appeared timely and relevant by taking into account the widespread dissemination of several rumors and unsubstantiated claims regarding HMPX [35,36]. These rumors that were shared widely on social media involved linking the occurrence of HMPX to COVID-19 vaccines, despite the absence of any evidence backing such claims, with fear of governmental restriction or lockdowns [62]. Conspiratorial beliefs regarding HMPX cannot be considered a novel phenomenon since it has been reported previously in endemic regions for the virus. For example, a previous study in the Republic of the Congo reported the endorsement of false notions, including the belief that the virus was deliberately introduced into the area and disbelief in the existence of disease [63].

The main finding of this study was the generally unsatisfactory levels of knowledge regarding monkeypox among university students in Jordanian health schools/faculties. University students in health schools/faculties are presumed to be a knowledgeable group, particularly in health-related topics [64]. One possible explanation for such gaps in knowledge of this emerging issue can be the poor coverage of the emerging viral infections, including monkeypox, in curricula of health schools in the country. A similar observation was reported by Harapan et al. in a recent study among the general practitioners in Indonesia [53].

Specific areas where a lack of knowledge can have significant negative public health consequences include the finding that only 48% of the medical students knew that antibiotics are not used to treat HMPX. This issue represents a major concern in Jordan, where the prevalence of utilizing antibiotics as a self-medication is prevalent at an alarming level [65,66,67]. Another important result is the general lack of knowledge of the utility of smallpox vaccination in the prevention of HMPX. In this study, only 26% of the respondents correctly knew that vaccination is available to prevent HMPX. The percentage was even lower among medical students, at 23%. As future healthcare workers, university students in health schools should be aware that they can provide cues to action through providing public health recommendations [68]. Lack of disease knowledge can negatively influence the recommendations of vaccine acceptance and adherence to public health intervention measures.

Variability in the per-item level of knowledge was noticed for various items between medical and non-medical students. However, medical students showed a significantly better level of knowledge only for the item “antibiotics are used to treat human monkeypox”. The generally lower levels of HMPX knowledge among medical students compared to their counterparts can be related to the confounding effect of a younger mean age among this group, with older age being correlated with a higher level of knowledge regarding HMPX. The aforementioned variability between medical and non-medical students was not seen upon comparing the overall level of knowledge based on the MPX K-score.

The assessment of HMPX knowledge among respondents in this study should be interpreted in relation to the timing of the survey. Since the start of May 2022, media coverage of HMPX has intensified in light of the increase in the number of cases during the current outbreak [69]. Thus, the reported level of knowledge among the students might not genuinely reflect the baseline level of educational material provided through university courses.

Emphasis on the importance of improving knowledge regarding HMPX should be highlighted due to the potential role of this approach in disease prevention [70]. This can be related to the nature of MPXV infection, where the animal reservoirs entail that eradication would be difficult and preventive measures including improved knowledge of the disease can be of utmost value [71].

The importance of improving students’ knowledge regarding the topics of emerging virus infections ushers the second important finding in this study; namely, the endorsement of conspiracy beliefs regarding virus emergence. In this study, the level of knowledge regarding HMPX was independently and significantly correlated with conspiracy beliefs about emerging virus infections. Thus, the integration of topics tackling the concepts of emerging infectious diseases in the curricula of medical and other health-related faculties/schools with innovative educational technologies can be beneficial for this aim [72]. This approach entails providing scientific and evidence-based explanations of the natural occurrence of emerging infectious diseases with insights into the potential increased frequency of its occurrence due to changes in the human behavior and ecological factors, among others [73,74]. Additionally, the improvement in the knowledge level of emerging infections can be beneficial in increasing levels of confidence in medical practice, as reported recently by Harapan et al. [75]. Furthermore, these educational improvements can be beneficial in enhancing the involvement of healthcare workers in preparedness for outbreaks and pandemics [76].

Besides the lower level of HMPX knowledge, female sex and affiliation to non-medical health schools/faculties were associated with a higher embrace of conspiracy beliefs regarding virus emergence in this study. A similar pattern was observed in our previous studies that investigated COVID-19 conspiracies among university students [44,48]. Despite the previous evidence that females were more likely to embrace conspiratorial ideas, especially in the studies conducted during the COVID-19 pandemic, as reviewed comprehensively by Valerie van Mulukom et al., more studies are needed to unveil the roots of associations between variables such as age, sex, educational level, etc., and the adoption of these harmful beliefs [45,77]. This is related to the reporting of conflicting results, with a few studies showing a lack of an association between sex and COVID-19 conspiracy beliefs and a study showing a higher likelihood of endorsing COVID-19 conspiracies among males [54,78,79]. The importance of unravelling predictors of conspiracy theories in the context of emerging virus infections is related to its severe negative consequences on health-related measures and less trust in science [80,81].

In this study, the finding of high levels of endorsement of conspiracy beliefs about emerging virus infection was an anticipated outcome. In the current milieu of the continuous emergence of infectious diseases, the embrace of conspiratorial explanations escalates due to fear and uncertainty [82]. In our previous study among the general population residing in Jordan, 57% of the study participants believed that SARS-CoV-2 origin was related to biological warfare [47]. In this study, and despite the inclusion of university students in health schools who are presumably more knowledgeable regarding health-related topics, about half of the study sample believed that MPXV origin is related to biological warfare. The negative impact of holding conspiracy beliefs has also been linked to higher anxiety levels, and higher prevalence of COVID-19 vaccine hesitancy, highlighting the importance of providing accurate information regarding emerging infectious diseases [44,48,49].

At least in simulation studies, the positive impact of mitigating misinformation in outbreak situations has been shown to improve the public health outcomes [83]. The belief in the role of biological warfare in relation to the spread of monkeypox is not a novel result of this study since it has been reported previously in Nigeria [84]. The government’s role in debunking such unsubstantiated notions was important.

The important role of scientific experts providing accurate and timely information about infectious disease outbreaks and their origin should be highlighted. In this study, the vast majority of respondents reported trusting information about the viruses from scientific experts. This result coupled with findings from recent studies on the role of scientific experts, physicians, and scientific journals highlights the significance of reliable sources of information in providing accurate knowledge, with a possible positive impact on health behavior [44,49,85].

### Strengths and Limitations

The strength of the current study is related to being among the first studies to correlate knowledge of the emerging monkeypox outbreak with conspiracy beliefs about virus emergence. Thus, the results of this study can be helpful to tailor well-informed educational and awareness programs and courses aiming to improve the knowledge on virus emergence. Subsequently, this can have a positive impact on health behavior of the future healthcare workers. Future research is recommended to evaluate the possible correlation between sources of information regarding emerging virus infections and the embrace of conspiracy beliefs.

The results of the study should be interpreted in light of several limitations, that included: (1) Sampling error, which was an obvious limitation of the study especially in relation to a majority of medical students and a majority of females; however, we believe that this effect is minimal based on the previous evidence of female predominance among students studying Dentistry, Nursing, and Pharmacy in Jordan [86,87]. (2) The relatively small sample size can also affect the generalizability of our results, particularly for non-medical students considering the extremely low number of Nursing, Pharmacy, Laboratory Sciences, and Rehabilitation students, besides the use of an electronic survey with a convenience sample. (3) The lack of survey items assessing the exact source of information about the disease was an important limitation which should be considered in the future studies.

## 5. Conclusions

Unsatisfactory levels of knowledge regarding HMPX were found among university students in this study. This appeared as an expected outcome considering the rarity of emerging virus infections’ coverage in curricula of health schools in the country. The high levels of endorsement of conspiratorial beliefs regarding emerging viral infections appeared as a recurring finding following the COVID-19 pandemic. The negative consequences of these beliefs necessitate proper intervention measures and should be considered in university educational courses and curricula. Further knowledge, attitude, and practices research among healthcare workers and the general population is also recommended and should be prioritized to fill gaps in knowledge regarding emerging zoonotic viral infections, including HMPX.

## Figures and Tables

**Figure 1 medicina-58-00924-f001:**
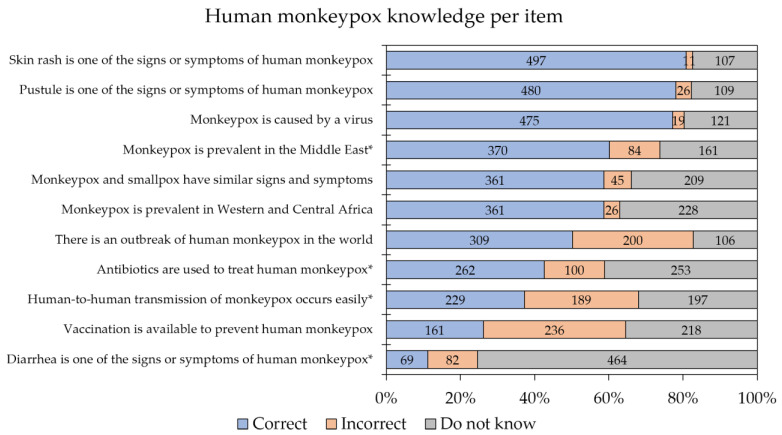
The overall level of human monkeypox knowledge among the study respondents. Human monkeypox knowledge items that are marked with an asterisk represent incorrect statements.

**Figure 2 medicina-58-00924-f002:**
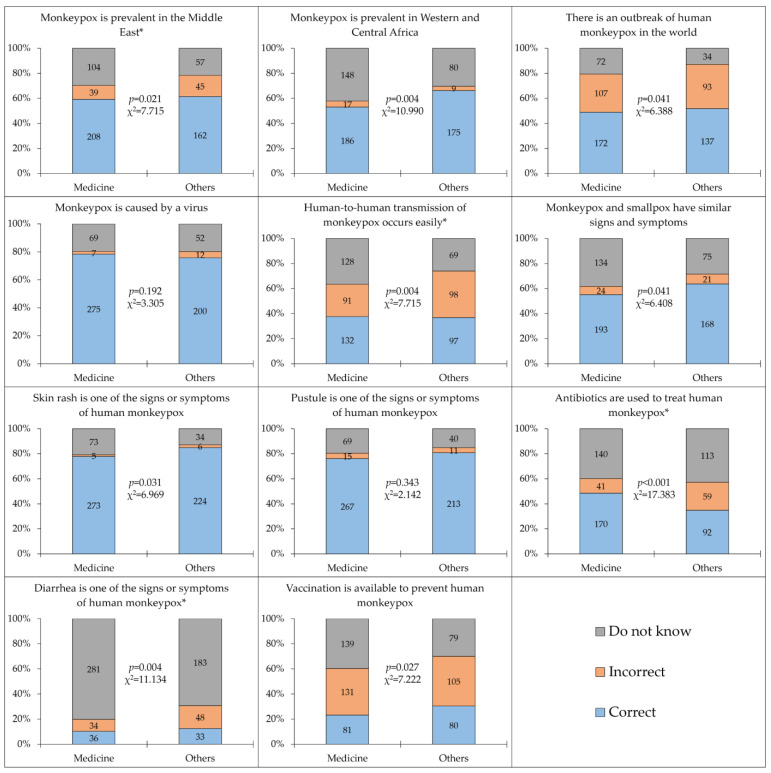
The level of human monkeypox knowledge among the study respondents divided by the school/faculty. Others: Non-medical schools, including Dental, Nursing, Pharmacy, Laboratory Sciences, and Rehabilitation schools. Human monkeypox knowledge items that are marked with an asterisk represent incorrect statements.

**Figure 3 medicina-58-00924-f003:**
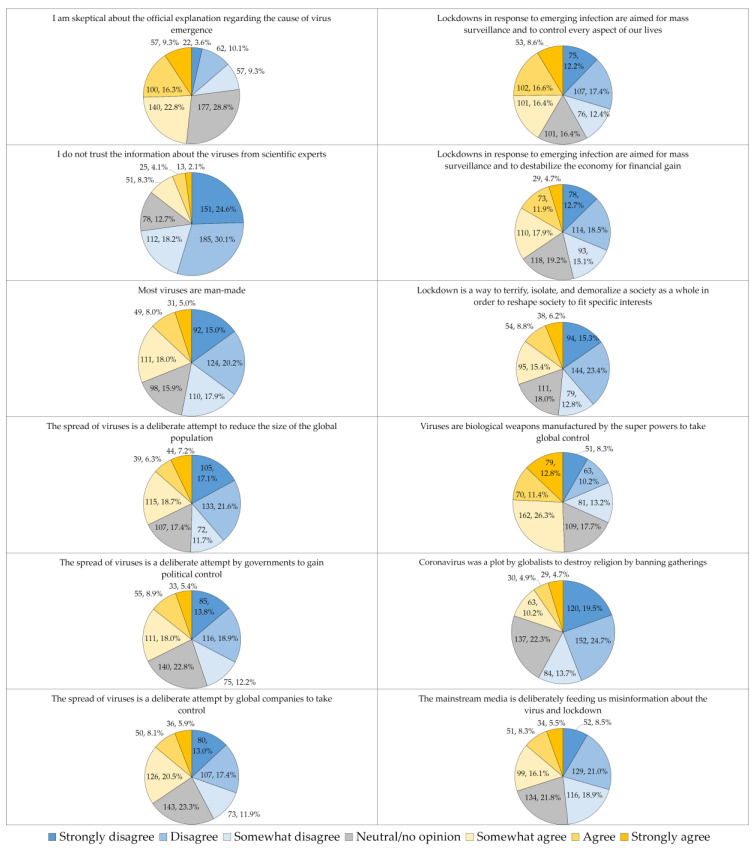
The attitude towards emerging virus infections conspiracy beliefs items in the whole study sample.

**Figure 4 medicina-58-00924-f004:**
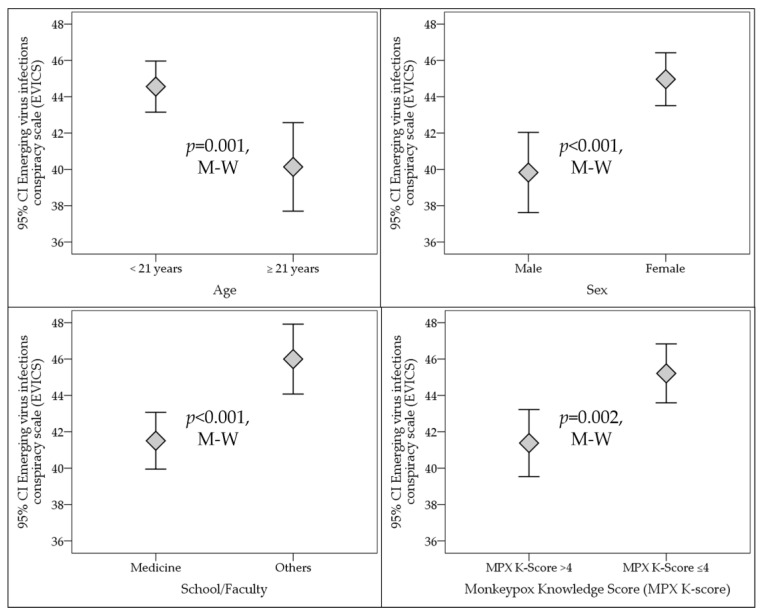
Factors that were significantly correlated with emerging virus infections conspiracy beliefs among the study participants. Others: Non-medical schools, including Dental, Nursing, Pharmacy, Laboratory Sciences, and Rehabilitation schools. MPX K-score: Human monkeypox knowledge score. M-W: Mann–Whitney *U* test.

**Table 1 medicina-58-00924-t001:** General characteristics of the study respondents (*n* = 615).

Variable	Group	Number (%)
Age in years (mean, SD ^1^)		19.9 (1.4)
Age	<21 years	459 (74.6)
	≥21 years	156 (25.4)
Sex	Male	183 (29.8)
	Female	432 (70.2)
Place of residence	Amman (the capital)	503 (81.8)
	Outside Amman	112 (18.2)
School/Faculty	Medicine	351 (57.1)
	Dentistry	191 (31.1)
	Nursing	12 (2.0)
	Pharmacy	23 (3.7)
	Laboratory Sciences	31 (5.0)
	Rehabilitation	7 (1.1)

^1^ SD: Standard deviation.

**Table 2 medicina-58-00924-t002:** The level of human monkeypox knowledge among the study respondents divided by age and sex.

Human Monkeypox Knowledge Item	Response	Age		*p*-Value, χ^2^	Sex		*p*-Value, χ^2^
		<21 years	≥21 years		Male	Female	
Monkeypox is prevalent in the Middle East *	Correct	272 (59.3)	98 (62.8)	0.357, 2.061	109 (59.6)	261 (60.4)	0.769, 0.527
	Incorrect	68 (14.8)	16 (10.3)		23 (12.6)	61 (14.1)	
	Do not know	119 (25.9)	42 (26.9)		51 (27.9)	110 (25.5)	
Monkeypox is prevalent in Western and Central Africa	Correct	257 (56.0)	104 (66.7)	0.064, 5.497	91 (49.7)	270 (62.5)	0.001, 15.040
	Incorrect	21 (4.6)	5 (3.2)		15 (8.2)	11 (2.5)	
	Do not know	181 (39.4)	47 (30.1)		77 (42.1)	151 (35.0)	
There is an outbreak of human monkeypox in the world	Correct	238 (51.9)	71 (45.5)	0.121, 4.225	100 (54.6)	209 (48.4)	0.364, 2.023
	Incorrect	150 (32.7)	50 (32.1)		54 (29.5)	146 (33.8)	
	Do not know	71 (15.5)	35 (22.4)		29	77	
Monkeypox is caused by a virus	Correct	339 (73.9)	136 (87.2)	0.003, 11.781	143 (78.1)	332 (76.9)	0.897, 0.217
	Incorrect	16 (3.5)	3 (1.9)		6 (3.3)	13 (3.0)	
	Do not know	104 (22.7)	17 (10.9)		34 (18.6)	87 (20.1)	
Human-to-human transmission of monkeypox occurs easily *	Correct	170 (37.0)	59 (37.8)	0.430, 1.690	75 (41.0)	154 (35.6)	0.141, 3.914
	Incorrect	136 (29.6)	53 (34.0)		46 (25.1)	143 (33.1)	
	Do not know	153 (33.3)	44 (28.2)		62 (33.9)	135 (31.3)	
Monkeypox and smallpox have similar signs and symptoms	Correct	254 (55.3)	107 (68.6)	<0.001, 16.076	98 (53.6)	263 (60.9)	0.189, 3.334
	Incorrect	29 (6.3)	16 (10.3)		17 (9.3)	28 (6.5)	
	Do not know	176 (38.3)	33 (21.2)		68 (37.2)	141 (32.6)	
Skin rash is one of the signs or symptoms of human monkeypox	Correct	359 (78.2)	138 (88.5)	0.019, 7.941	140 (76.5)	357 (82.6)	0.210, 3.122
	Incorrect	9 (2.0)	2 (1.3)		4 (2.2)	7 (1.6)	
	Do not know	91 (19.8)	16 (10.3)		39 (21.3)	68 (15.7)	
Pustule is one of the signs or symptoms of human monkeypox	Correct	347 (75.6)	133 (85.3)	0.041, 6.384	144 (78.7)	336 (77.8)	0.473, 1.499
	Incorrect	22 (4.8)	4 (2.6)		5 (2.7)	21 (4.9)	
	Do not know	90 (19.6)	19 (12.2)		34 (18.6)	75 (17.4)	
Antibiotics are used to treat human monkeypox *	Correct	174 (37.9)	88 (56.4)	<0.001, 17.084	81 (44.3)	181 (41.9)	0.653, 0.853
	Incorrect	84 (18.3)	16 (10.3)		26 (14.2)	74 (17.1)	
	Do not know	201 (43.8)	52 (33.3)		76 (41.5)	177 (41.0)	
Diarrhea is one of the signs or symptoms of human monkeypox *	Correct	45 (9.8)	24 (15.4)	0.158, 3.692	14 (7.7)	55 (12.7)	0.047, 6.127
	Incorrect	63 (13.7)	19 (12.2)		19 (10.4)	63 (14.6)	
	Do not know	351 (76.5)	113 (72.4)		150 (82.0)	314 (72.7)	
Vaccination is available to prevent human monkeypox	Correct	109 (23.7)	52 (33.3)	<0.001, 15.938	52 (28.4)	109 (25.2)	0.430, 1.689
	Incorrect	167 (36.4)	69 (44.2)		73 (39.9)	163 (37.7)	
	Do not know	183 (39.9)	35 (22.4)		58 (31.7)	160 (37.0)	

* Human monkeypox knowledge items that are marked with an asterisk represent incorrect statements.

## Data Availability

The data presented in this study are available upon request from the corresponding author (M.S.).

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
