# Peer review of "Knowledge of Human Monkeypox and Its Relation to Conspiracy Beliefs among Students in Jordanian Health Schools: Filling the Knowledge Gap on Emerging Zoonotic Viruses"

_medicina, 2022, doi:10.3390/medicina58070924_

Round 1
Reviewer 1 Report
This manuscript evaluates the level of knowledge of human monkeypox (HMPX)and its relation to conspiracy beliefs among university students in Jordanian health schools in order to fill the knowledge gap on emerging zoonotic viruses. Authors contend that HMPX poses an emerging public health concern. Moreover, university students in health schools as future healthcare workers should be equipped with accurate knowledge about HMPX to guard against detrimental conspiracy beliefs.
The review is as follows:
1. The introduction is insightful and compelling.
2. There should be mention of the recent announcement by the World Health Organization (WHO) that it will rename the monkeypox virus after a group of scientists voiced concerns that the name is discriminatory and stigmatizing.
3. Line 97 – Define the word ‘clade’ for the lay reader.
4. Line 127 – Check capitalization in ‘The Overarching’.
5. Line 144 – In Materials and Methods, explain what is meant by ‘chain-referral sampling’.
6. Regarding data analysis, were there quality checks to ensure surveys were completed in entirety?
7. Lines 256-258 – In “Non- medical students had higher level of knowledge compared to medical students for eight items, compared to a single item where medical students displayed significantly higher level of knowledge”, these are intriguing findings. What explains this finding.
8. Lines 345-347- In “In this study, only 26% of the respondents correctly knew that vaccination is available to prevent HMPX. The percentage was even lower among medical students at 23%”. This is intriguing that the knowledge was lower among medical students. Authors should expand on this and discuss possible explanations.
Overall, this is a pertinent article on a unique topic. It is compelling and interesting to read. Authors should discuss current WHO recommendations to rename monkeypox due to stigma concerns. Expand discussion on the findings related to lower HMPX knowledge among medical students. Address some clarifying questions on the materials, methods, and data analysis. Addressing these items should help improve the paper.
Author Response
Reviewer #1 Comments and Suggestions for Authors
This manuscript evaluates the level of knowledge of human monkeypox (HMPX)and its relation to conspiracy beliefs among university students in Jordanian health schools in order to fill the knowledge gap on emerging zoonotic viruses. Authors contend that HMPX poses an emerging public health concern. Moreover, university students in health schools as future healthcare workers should be equipped with accurate knowledge about HMPX to guard against detrimental conspiracy beliefs.
Response: We are deeply thankful for the insightful summary and for the positive critical appraisal of the manuscript by the estimated reviewer.
The review is as follows:
- The introduction is insightful and compelling.
Response: We are deeply thankful for the this note.
- There should be mention of the recent announcement by the World Health Organization (WHO) that it will rename the monkeypox virus after a group of scientists voiced concerns that the name is discriminatory and stigmatizing.
Response: We would like to thank the reviewer for this important and timely point. We totally agree that the nomenclature scheme for human monkeypox virus clades should be changes into non-discriminatory and non-stigmatizing description following the proposal by Christian Happi et al in: https://virological.org/t/urgent-need-for-a-non-discriminatory-and-non-stigmatizing-nomenclature-for-monkeypox-virus/853
Accordingly, and based on the important suggestion by the reviewer, we added the following paragraph in the Introduction section (Page 3, lines 105-118 of the revised highlighted manuscript):
“An important point to be emphasized is the urgent need to adopt a new non-discriminatory and non-stigmatizing nomenclature scheme for MPXV clades [31]. One proposed classification scheme was conceptualized by Christian Happi et al. based on maximum likelihood phylogenetic analysis of the available MPXV genomes (1970-2022). This “Happi” classification system suggested the adoption of Arabic numerals to assign MPXV clades based on their order of detection rather than source of isolation [31]. Thus, the older nomenclature will be rendered obsolete by the use of “MPXV clade 1” instead of Congo Basin clade, “MPXV clades 2 and 3” instead of West African clade, with the newly assigned clade 3 incorporating most genomes from the human outbreaks that were recorded in 2017, 2018 and the ongoing 2022 multi-country outbreak. Additionally, the proposed “Happi” scheme inferred the discernible genetic diversity of the taxa within clade 3; therefore, the adoption of the “Pango” nomenclature scheme used for SARS-CoV-2 was suggested for lineages within this clade (e.g. A.1, A.2, A.1.1, and B.1 representing the taxa that were sequenced during the ongoing multi-country HMPX outbreak) [31,32].”
The following references were added to support the statements mentioned in the newly added paragraph:
- Happi, C.; Adetifa, I.; Mbala, P.; Njouom, R.; Nakoune, E.; Happi, A.; Ndodo, N.; Ayansola, O.; Mboowa, G.; Bedford, T., et al. Urgent need for a non-discriminatory and non-stigmatizing nomenclature for monkeypox virus. Availabe online: https://virological.org/t/urgent-need-for-a-non-discriminatory-and-non-stigmatizing-nomenclature-for-monkeypox-virus/853 (accessed on 19 June 2022).
- Rambaut, A.; Holmes, E.C.; O’Toole, Á.; Hill, V.; McCrone, J.T.; Ruis, C.; du Plessis, L.; Pybus, O.G. A dynamic nomenclature proposal for SARS-CoV-2 lineages to assist genomic epidemiology. Nat Microbiol 2020, 5,(11): 1403-1407, doi:10.1038/s41564-020-0770-5.
- Line 97 – Define the word ‘clade’ for the lay reader.
Response: Based on the reviewer’s important suggestion, we added the definition of clades to the Introduction section as follows (Page 2, line 97 of the revised highlighted manuscript):
“The epidemiology of MPXV infection has been dominated by two phylogenetically distinct clades (monophyletic taxa descending from a common ancestor) of the virus”.
- Line 127 – Check capitalization in ‘The Overarching’.
Response: We are thankful for this note, and accordingly we corrected this typographical error.
- Line 144 – In Materials and Methods, explain what is meant by ‘chain-referral sampling’.
Response: Based on the reviewer’s comment, we added the following statement to the Materials and Methods section (Page 4, lines 161-164 of the revised highlighted manuscript):
“Recruitment of the potential respondents was done through chain-referral sampling, starting by the contacts of the authors (four of whom are instructors of medical, dental, nursing and medical laboratory sciences students, and five medical/dental students at the University of Jordan) with reliance on participants’ referral of the survey link to their contacts [48].” Additionally, we added the following reference to support the newly added paragraph:
- Penrod, J.; Preston, D.B.; Cain, R.E.; Starks, M.T. A Discussion of Chain Referral As a Method of Sampling Hard-to-Reach Populations. J Transcult Nurs 2003, 14,(2): 100-107, doi:10.1177/1043659602250614.
- Regarding data analysis, were there quality checks to ensure surveys were completed in entirety?
Response: We are deeply thankful for this meticulous note. However, we ensured that surveys were completed in entirety by making the survey items mandatory. Thus, submission of the responses was not allowed unless the respondents answered the entire questionnaire. This was done to eliminate the potential item non-response issue. This was previously mentioned in the Materials and Methods section: “Response to all items was mandatory to overcome the item non-response issue.” Page 4, line 168 of the revised highlighted manuscript.
- Lines 256-258 – In “Non- medical students had higher level of knowledge compared to medical students for eight items, compared to a single item where medical students displayed significantly higher level of knowledge”, these are intriguing findings. What explains this finding.
Response: We are thankful for this important note, which highlighted this interesting result. As mentioned in the Discussion section (Page 12, lines 393-398 of the revised highlighted manuscript), we sought to find explanations for this interesting finding and the plausible reason for such a discrepancy in the level of knowledge with higher levels observed among non-medical students could have been related to the age variable. Since older age was found to be correlated with higher HMPX levels of knowledge for a majority of items, and the mean age of medical students was significantly younger than non-medical students, we are inclined to believe that age confounded this result. This issue was highlighted in the Discussion section: “Variability in the per-item level of knowledge was noticed for various items between medical and non-medical students. However, medical students showed significantly better level of knowledge only for the item “antibiotics are used to treat human monkeypox”. The generally lower levels of HMPX knowledge among medical students compared to their counterparts can be related to the confounding effect of younger mean age among this group, with older age being correlated with a higher level of knowledge regarding HMPX. The aforementioned variability between medical and non-medical students was not seen upon comparing the overall level of knowledge based on the MPX K-score.”
In addition, the general level of knowledge between the two groups (medical vs. non-medical students) as evaluated based on the overall MPX knowledge score did not yield any statistically significant differences. Based on the reviewer comment, we added the following paragraph to the Results section (Page 7, lines 285-287 of the revised highlighted manuscript):
“However, the assessment of the difference between the mean MPX K-score among medical vs. non-medical students did not yield a statistically significant difference (4.2 vs. 4.1 respectively, p=0.523, M-W).”
Additionally, we added the following paragraph to the Discussion section (Page 12, line 399-400):
“The aforementioned variability between medical and non-medical students was not seen upon comparing the overall level of knowledge based on the MPX K-score.”
- Lines 345-347- In “In this study, only 26% of the respondents correctly knew that vaccination is available to prevent HMPX. The percentage was even lower among medical students at 23%”. This is intriguing that the knowledge was lower among medical students. Authors should expand on this and discuss possible explanations.
Response: Again, we are thankful for this important point raised by the reviewer; however, we believe that our explanation in the previous point #7 addressed this issue (age-related variability).
Overall, this is a pertinent article on a unique topic. It is compelling and interesting to read. Authors should discuss current WHO recommendations to rename monkeypox due to stigma concerns. Expand discussion on the findings related to lower HMPX knowledge among medical students. Address some clarifying questions on the materials, methods, and data analysis. Addressing these items should help improve the paper.
Response: We are deeply grateful for the comprehensive and detailed reviewer’s comments that helped us to clarify many issues in the manuscript. Thank you!
Reviewer 2 Report
I have reviewed the paper by Sallam et al. on knowledge of human monkeypox by health schools students in Jordan.
The article is relevant in the context of knowledge being key to preparedness for fighting pandemics, and to counteract the risk of misinformation.
More than half of the responders where medical students, and they had a lower mean age. We need to details on where the majority of the female responders came from, as one 2% were nursing students. So far, all we know is that “the percentage of male medical students was significantly higher compared to the 243 percentage of male respondents from other schools/faculties.
I think it makes sense to compare responses from medical students vs. other schools (Figure 2). Table 3 results only matter if we knew where the female responders are coming from.
Figure 3 is not useful and could be placed as Supplementary material.
Figure 4. is crucial and needs to be discussed extensively.
Author Response
Reviewer #2 Comments and Suggestions for Authors
I have reviewed the paper by Sallam et al. on knowledge of human monkeypox by health schools students in Jordan.
The article is relevant in the context of knowledge being key to preparedness for fighting pandemics, and to counteract the risk of misinformation.
Response: We are deeply thankful for the positive critical appraisal of the manuscript.
More than half of the responders where medical students, and they had a lower mean age. We need to details on where the majority of the female responders came from, as one 2% were nursing students. So far, all we know is that “the percentage of male medical students was significantly higher compared to the 243 percentage of male respondents from other schools/faculties.
Response: We are grateful for this important comment that needs more elaboration. In Jordan, female predominance is seen in the following university health schools/faculties: Dentistry, Nursing, Rehabilitation and Medical Laboratory Sciences with about (60-90%) female students, while males predominate in other schools (e.g. engineering, physical education). For medical schools, slight female predominance is seen as well; however, this female predominance is less conspicuous. This predominance has been demonstrated previously in different KAP studies as follows:
- Sallam, M.; Al-Fraihat, E.; Dababseh, D.; Yaseen, A.; Taim, D.; Zabadi, S.; Hamdan, A.A.; Hassona, Y.; Mahafzah, A.; Åžahin, G.Ö. Dental students’ awareness and attitudes toward HPV-related oral cancer: a cross sectional study at the University of Jordan. BMC Oral Health 2019, 19,(1): 171, doi:10.1186/s12903-019-0864-8.
- Abuhammad, S.; Muflih, S.; Alzoubi, K.H.; Gharaibeh, B. Nursing and PharmD Undergraduate Students' Attitude Toward the "Do Not Resuscitate" Order for Children with Terminally Ill Diseases. J Multidiscip Healthc 2021, 14: 425-434, doi:10.2147/JMDH.S298384.
However, we agree that the extremely low number of students from nursing, pharmacy, laboratory sciences and rehabilitation students. Therefore, and in response to the important comment by the reviewer, we added the following paragraph to the Results section (Page 6, lines 255-258 of the revised highlighted manuscript):
“Medical and dental students prevailed in the study sample, while female students represented the majority of respondents across all schools/faculties (63.2% of medical students, 75.0% of nursing students, 78.0% of dental students, 82.6% of pharmacy students, 85.7% of rehabilitation students and 87.1% of laboratory sciences students).”
Also, we added the following paragraph to the limitations sub-section of the Discussion (Page 14, lines 477-479):
“The relatively small sample size can also affect the generalizability of our results particularly for non-medical students considering the extremely low number of nursing, pharmacy, laboratory sciences and rehabilitation students”
I think it makes sense to compare responses from medical students vs. other schools (Figure 2). Table 3 results only matter if we knew where the female responders are coming from.
Response: We would like to thank the reviewer for this point. However, we believe that showing the significant per-item variability was not reflected on the overall MPX K-score. Therefore, we added this paragraph to the Results section (Page 7, lines 287-288 of the revised highlighted manuscript):
“The mean MPX K-score was similar upon stratification per sex (4.2 for both male and female students, p=0.889, M-W).”
Figure 3 is not useful and could be placed as Supplementary material.
Response: We would like to thank the reviewer for this suggestion; however, we are inclined to keep this Figure in the manuscript since it gives a general idea regarding the embrace of general conspiracy beliefs towards virus emergence and its depicts the differences observed per item.
Figure 4. is crucial and needs to be discussed extensively.
Response: We would like to thank the reviewer for this comment, and accordingly we amended the Discussion section as follows (Page 13, lines 426-439 of the revised highlighted manuscript):
“Besides the lower level of HMPX knowledge, female sex and affiliation to non-medical health schools/faculties were associated with higher embrace of conspiracy beliefs regarding virus emergence in this study. Similar pattern was observed in our previous studies that investigated COVID-19 conspiracies among university students [44,48]. Despite the previous evidence that females were more likely to embrace conspiratorial ideas, especially in the studies conducted during the COVID-19 pandemic as reviewed comprehensively by Valerie van Mulukom et al., more studies are needed to unveil the roots of associations between variables like age, sex, educational level, etc. and adoption of these harmful beliefs [45,77]. This is related to reporting of conflicting results, with a few studies showing lack of an association between sex and COVID-19 conspiracy beliefs and a study showing higher likelihood of endorsing COVID-19 conspiracies among males [54,78,79]. The importance of unravelling predictors of conspiracy theories in the context of emerging virus infections is related to its severe negative consequences on health-related measures and less trust in science [80,81].”
We would like to thank the reviewer for the important suggestions and remarks.